# CLED-Fusion: Controllable and Latent-Explainable Diffusion for Multi-Degradation Multi-Modal Image Fusion

## Abstract

Multi-modal image fusion aims to combine complementary information from different modalities, yet its deployment is hindered by diverse degradations (*e.g.*, low-light, blur, haze, noise). Existing methods mainly focus on feature integration, lacking controllability over degradation removal and explainability of the generative process. We propose a novel **C**ontrollable and **L**atent-**E**xplainable **D**iffusion framework for multi-degradation **Fusion** (**CLED-Fusion**). CLED-Fusion introduces shared distribution priors to unify heterogeneous degradations into a consistent latent space, enabling controllable regulation of removal strength and cross-modal balance. Diffusion dynamics are reformulated into a dual process, where a deterministic residual pathway removes degradations and a stochastic noise pathway preserves fine details, yielding an interpretable trajectory. An explicit degrade-fusion module embeds these priors directly into degraded inputs, avoiding redundant reconstruction and ensuring efficiency. Extensive experiments on multiple benchmarks show that CLED-Fusion achieves superior fusion quality, robustness to degradations, and strong adaptability in medical imaging scenarios. The code is available at `https://anonymous.4open.science/r/CLED-Fusion-D88C/`.

## 1 Introduction

Multi-modal image fusion aims to integrate information from multiple images of the same scene or object, acquired from different sensors, into a single composite image Wu et al. (2025); Wang et al. (2025a). The resulting fused image typically offers more comprehensive and accurate information, facilitating easier comprehension and analysis by human observers Liu et al. (2024); Wang et al. (2020); Li et al. (2024). Furthermore, it provides richer and more robust features for various image processing tasks Luo et al. (2019); Jin et al. (2025), thereby significantly enhancing the accuracy and performance of downstream applications. Currently, image fusion technology finds widespread applications in diverse fields, *e.g.*, autonomous driving Hu et al. (2024), medical imaging Basu et al. (2024) and military security Sun et al. (2024).

Propelled by rapid advancements in deep learning Luo et al. (2021); Wang et al. (2023); Li et al. (2025), the field of image fusion has witnessed significant breakthroughs in recent years. Unlike traditional methods, deep learning-based approaches demonstrate immense potential by leveraging data-driven paradigms to yield superior fusion quality with minimal manual feature engineering Zhang & Demiris (2023). Architecturally, these methods encompass diverse deep network architectures, *e.g.*, convolutional neural networks (CNN) Tang et al. (2022b), autoencoders (AE) Li & Wu (2018), generative adversarial networks Ma et al. (2019), Transformers Zhao et al. (2023a), and diffusion models. Notably, their development has closely mirrored the broader evolution of neural network paradigms. This strong correlation is primarily attributable to the inherent scarcity of ground truth data in image fusion, particularly in multi-modal contexts critically requiring robust feature extraction capabilities for superior fusion performance Wang et al. (2024b; 2022); Lou et al. (2025).

However, real-world environments present significant challenges to image acquisition, introducing various degradations, *e.g.*, low-light, blur, overexposure, and noise due to diverse weather condi-

tions and varying device performance Cao et al. (2025). Given the strong application-driven nature of image fusion, it must inherently address these complex degradation issues. Nevertheless, most current fusion methods exhibit suboptimal performance in such degraded scenarios. This limitation primarily arises because existing approaches predominantly focus on information integration rather than comprehensively addressing degraded image restoration. Furthermore, while some methods mitigate degradation interference to a certain extent, they often consider only a single type of degradation, for instance, exclusively tackling low-light conditions in visible images *e.g.*, DIVFusion Tang et al. (2023). Finally, even those methods attempting multi-degradation often lack scalability concerning the types of degradations during training, making it considerably challenging to incorporate additional degradation types into existing frameworks, *e.g.*, DRMF Tang et al. (2024) and Text-DiFuse Zhang et al. (2024). More importantly, current approaches largely overlook the *controllability* and *explainability* in the image fusion.

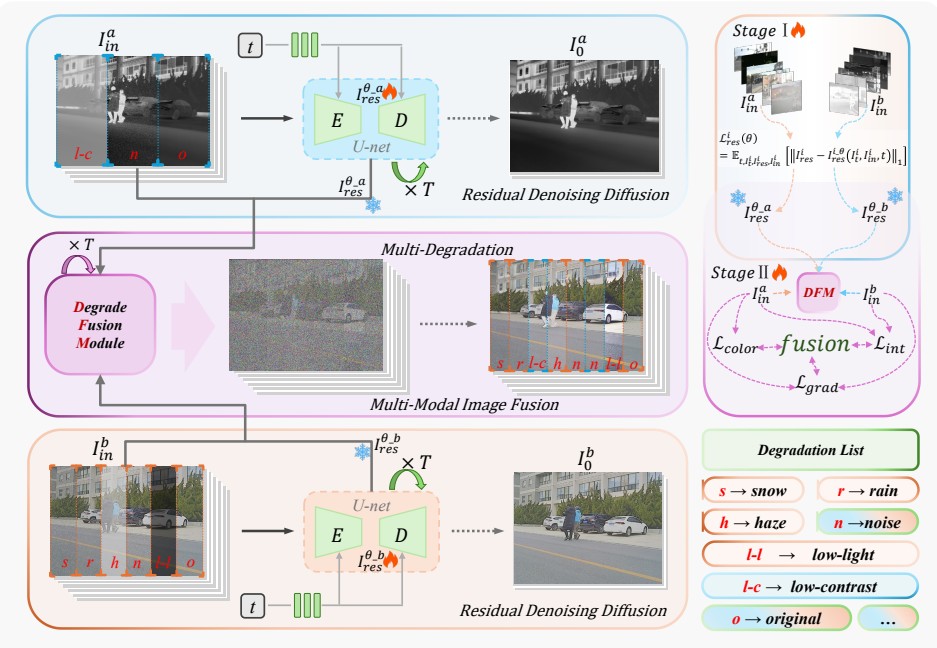

Figure 1: Overview of CLED-Fusion. CLED-Fusion first disentangles degradations from each modality via residual denoising diffusion, then employs a degrade-fusion module to integrate multi-degradation distributions into a unified latent space guided by shared priors, enabling controllable degradation removal and cross-modal semantic preservation.

To address these challenges, we propose **a Controllable and Latent-Explainable Diffusion (CLED)** framework for multi-degradation multi-modal image fusion as shown in Fig. 1. The framework involves two fundamental principles: *explainability* and *controllability*. For explainability, we reformulate the diffusion dynamics as a dual process composed of a deterministic residual pathway, explicitly capturing degradation factors such as blur, low-light, or haze, and a stochastic noise pathway to model uncertainty and fine-grained details. This disentangled design provides a transparent generative trajectory, where each reverse step can be interpreted as progressively removing degradation while retaining semantic information. For controllability, we propose shared distribution priors to unify heterogeneous degradation patterns across modalities into a consistent high-quality latent representation. Meanwhile, the learned priors enable precise regulation over the degree of degradation removal with an explicit fusion module, ensuring that the fusion results remain physically plausible while exploiting complementary information across modalities, and avoiding the superfluous reconstructing images from pure Gaussian noise as in conventional diffusion methods. By embedding diffusion priors into this structured process, the framework delivers semantically faithful, robust, and high-quality fusion results with only a few denoising steps.

In summary, we make the following contributions:

- We propose a novel CLED framework for multi-degradation multi-modal image fusion. The framework explicitly disentangles diffusion dynamics into a deterministic residual pathway, modeling degradation factors and a stochastic noise pathway to capture uncertainty and fine-grained details. The designed CLED provides a transparent and interpretable generative trajectory for multi-modal image fusion under complex degradations.

- To achieve controllability of image fusion, we design shared distribution priors to unify heterogeneous degradation patterns across modalities into a consistent high-quality latent space. The learned priors enable precise regulation of degradation removal strength and make the fusion results remain both physically plausible and semantically complementary across modalities.

- We further develop an explicit degrade-fusion module to leverage degraded inputs directly, where the diffusion priors are embedded into the fusion process to avoid redundant reconstruction from Gaussian noise and produces semantically faithful, robust, and high-quality fusion results with only a few denoising steps.

- Extensive experiments on diverse benchmarks demonstrate the superiority of CLED over state-of-the-art methods, highlighting its robustness to degradations, adaptability to downstream tasks, and effectiveness in medical multi-modal domain.

## 2 RELATED WORK

### 2.1 MULTI-MODAL IMAGE FUSION

Deep learning in image fusion has progressed from early autoencoder (AE)-based methods Li & Wu (2018); Wang et al. (2025b), which used hand-designed rules for feature integration but had performance limits. Subsequently, CNNs and Transformers Huang et al. (2022) enabled end-to-end fusion leveraging their respective local and global feature extraction capabilities and significantly improved performance via unsupervised loss functions Tang et al. (2022b); Kang et al. (2024); Wang et al. (2021); Qin et al. (2025). GANs then emerged offering unsupervised learning through adversarial training to preserve multi-modal features despite often facing training instability and mode collapse Ma et al. (2019). More recently, diffusion models have become a promising research direction Yi et al. (2024a); Wang et al. (2025c); Cai et al. (2025) due to their superior training stability, enhanced sample diversity, and ability to generate high-quality detailed images. Early works include DDFM Zhao et al. (2023b) by Zhao *et al.*, which sampled fused images from pre-trained diffusion models, and Dif-Fusion Yue et al. (2023) by Yue *et al.*, which directly generated chromatic fused images while ensuring fidelity in color, gradient, and intensity. Cao et al. (2024) also proposed CCF for controllable and dynamic image fusion. However, these initial methods did not fully exploit diffusion models' degradation removal capabilities. Consequently, Tang et al. (2024) and Zhang et al. (2024) further investigated using diffusion models' generative power for degradation issues. Nevertheless, the range of degradations these approaches could handle was design-dependent, limiting extensibility, and their degradation robustness was constrained by treating degraded images merely as sampling process constraints.

### 2.2 DIFFUSION MODEL

Diffusion models, which approximate target data distributions through learned processes, have garnered significant research attention recently due to their remarkable performance in generative tasks. Subsequently, works in image restoration, *e.g.*, Sun et al. (2023), SR3 Saharia et al. (2022), and WeatherDiffusion Özdenizci & Legenstein (2023), progressively harmonized their generative and restorative capabilities. This inherent capability naturally attracted interest and extension within the image fusion domain. However, conditioning the denoising network on degraded images and initiating the reverse process from pure noise appears both unnecessary and inefficient for image fusion tasks. Consequently, numerous recent studies have explored generating clear images directly from degraded or noise-corrupted degraded inputs. DRMF Tang et al. (2024) and Text-DiFuse Zhang et al. (2024) attempt to use diffusion model to handle multi-modal image fusion and obtain the excellent effect. Although effective, the challenges of achieving full controllability and explainability remain unresolved.

## 3 PROPOSED METHOD

### 3.1 UNIFIED MULTI-DEGRADATION DISTRIBUTION

#### 3.1.1 CONTROLLABLE RESIDUAL–NOISE DECOMPOSITION

Denoising diffusion models aim to approximate a target data distribution $q(I_0)$ with a learned distribution $p_\theta(I_0) = \int p_\theta(I_{0:T})dI_{1:T}$ Ho et al. (2020). In the forward process, $q(I_0)$ is diffused into a Gaussian noise distribution using a fixed Markov chain:

$$q(I_{1:T}|I_0) = \prod_{t=1}^T q(I_t|I_{t-1}), \tag{1}$$

$$q(I_t|I_{t-1}) = \mathcal{N}(I_t; \sqrt{\alpha_t}I_{t-1}, (1-\alpha_t)\mathbf{I}), \tag{2}$$

where $\alpha_{1:T} \in (0,1]^T$. $q(I_t|I_{t-1})$ can also be written as $I_t = \sqrt{\alpha_t}I_{t-1} + \sqrt{1-\alpha_t}\epsilon_{t-1}$. By reparameterization trick, $I_t$ can be directly sampled from $I_0$:

$$I_t = \sqrt{\bar{\alpha}_t}I_0 + \sqrt{1-\bar{\alpha}_t}\epsilon, \tag{3}$$

where $\epsilon \sim \mathcal{N}(\mathbf{0}, \mathbf{I}), \bar{\alpha}_t = \prod_{s=1}^t \alpha_s$.

While effective in generative modeling, this purely stochastic formulation lacks the ability to explicitly capture degradations and control their removal. To address this, we reformulate the forward process as a dual-pathway decomposition: (i) a deterministic residual pathway that explicitly models structured degradations (e.g., blur, low-light, haze), and (ii) a stochastic noise pathway that captures uncertainty and fine-grained details. Then, the forward step is expressed as:

$$I_t = I_{t-1} + I_{res}^t, \qquad I_{res}^t \sim \mathcal{N}(\alpha_t I_{res}, \beta_t^2 \mathbf{I}), \tag{4}$$

where $I_t$ denotes the diffusion state at step $t$, $I_{in}$ is the degraded input (e.g., noisy, low-light, or blurred), and $I_{res}$ is the residual between $I_{in}$ and the clean image $I_0$, i.e., $I_{res} = I_{in} - I_0$. Importantly, $\alpha_t$ and $\beta_t$ act as independent control coefficients, explicitly regulating the contribution of residual degradation and stochastic noise, thereby providing fine-grained controllability over the forward diffusion dynamics. Based on this formulation, the single-step noising process can be expressed as:

$$q(I_t|I_{t-1}, I_{res}) = \mathcal{N}(I_t; I_{t-1} + \alpha_t I_{res}, \beta_t^2 \mathbf{I}). \tag{5}$$

In the reverse process, $q(I_{t-1}|I_t, I_0^\theta, I_{res}^\theta)$ is used to simulate the true generative distribution $p_\theta(I_{t-1}|I_t)$, which can also be written as a Markov chain:

$$p_\theta(I_{t-1}|I_t) = \mathcal{N}(I_{t-1}; I_0^\theta + \overline{\alpha}_{t-1}I_{res}^\theta + \overline{\beta}_{t-1}\epsilon^\theta, \mathbf{0} \cdot \mathbf{I}), \tag{6}$$

where terms with $\theta$ indicate they are obtained from the model output. Since in the forward process, residuals ($I_{res}$) and noise ($\epsilon$) are gradually added to $I_0$ and then synthesized into $I_t$, the reverse process from $I_T$ to $I_0$ involves estimating the residuals and noise injected in the forward process. Therefore, it is necessary to train a residual network $I_{res}^\theta(I_t, t, I_{in})$ to predict $I_{res}$ and a noise network $\epsilon_\theta(I_t, t, I_{in})$ to estimate $\epsilon$. Thus, the estimated target image is $I_0^\theta = I_t - \bar{\alpha}_t I_{res}^\theta - \bar{\beta}_t \epsilon_\theta$. Given $I_0^\theta$ and $I_{res}^\theta$, the generation process is defined as:

$$I_0^\theta = I_t - \bar{\alpha}_t I_{res}^\theta - \bar{\beta}_t \epsilon_\theta, \qquad p_\theta(I_{t-1}|I_t) = q(I_{t-1}|I_t, I_0^\theta, I_{res}^\theta). \tag{7}$$

Correspondingly, the training loss function for this process includes two terms:

$$\mathcal{L}_{res}(\theta) = \mathbb{E}\left[\left\|I_{res} - I_{res}^\theta(I_t, I_{in}, t)\right\|^2\right], \tag{8}$$

$$\mathcal{L}_\epsilon(\theta) = \mathbb{E}\left[\left\|\epsilon - \epsilon_\theta(I_t, I_{in}, t)\right\|^2\right]. \tag{9}$$

This dual decomposition is the cornerstone of our framework, ensuring explainability by making each reverse step interpretable as degradation removal, and controllability by independently regulating residual and noise contributions.

### 3.1.2 SHARED-PRIOR UNIFIED MAPPING

Although residual–noise decomposition enhances interpretability, modality-specific degradations remain heterogeneous. To achieve cross-modal consistency, we introduce shared distribution priors to embed different degradations into a unified latent space. This not only ensures consistent fusion across modalities but also enables fine-grained control of degradation removal. Consequently, to facilitate shared distribution mapping, a selective hourglass mapping strategy based forward process is developed as:

$$q(I_t|I_{t-1}, I_{res}, I_{in}) = \mathcal{N}(I_t; I_{t-1} + \alpha_t I_{res} - \delta_t I_{in}, \beta_t^2 \mathbf{I}), \tag{10}$$

where $\delta_t I_{in}$ is the shared distribution term, and $\delta$ is the shared distribution coefficient.

We apply this idea to the image fusion problem, where the primary objective is to establish a Shared-Prior Unified Mapping that aligns multi-degradation inputs from different modalities into a consistent latent space. Accordingly, based on the Markov chain and reparameterization techniques, the forward process in our context is formulated as:

$$q(I_t^i|I_0^i, I_{res}^i, I_{in}^i) = \mathcal{N}(I_t^i; I_0^i + \bar{\alpha}_t I_{res}^i - \bar{\delta}_t I_{in}^i, \bar{\beta}_t^2 \mathbf{I}), \tag{11}$$

$$I_t^i = I_0^i + \bar{\alpha}_t I_{res}^i + \bar{\beta}_t \epsilon_t - \bar{\delta}_t I_{in}^i, \tag{12}$$

where $\bar{\alpha}_t = \sum_{j=1}^t \alpha_j$, $\bar{\beta}_t = \sqrt{\sum_{j=1}^t \beta_j^2}$, $\bar{\delta}_t = \sum_{j=1}^t \delta_j$, $i \in \{a, b\}$, with $a$ and $b$ representing different modalities, and $\epsilon_t \sim \mathcal{N}(\mathbf{0}, \mathbf{I})$. When $t \to T$, $\bar{\alpha}_T = 1$, $\bar{\delta}_T = 0.9$, thus the formula can be further rewritten as $I_T^i = (1 - \bar{\delta}_T)I_{in}^i + \bar{\beta}_T \epsilon_T = 0.1 I_{in}^i + \bar{\beta}_T \epsilon_T$.

In the reverse process, our goal is to restore high-quality images from noisy-carrying degraded images, calculated as follows:

$$p_\theta(I_{t-1}^i|I_t^i, I_{in}^i) = \mathcal{N}(I_{t-1}^i; u_\theta(I_t^i, I_{in}^i, t), \sigma_t^2 \mathbf{I}), \tag{13}$$

where the mean $u_\theta(I_t^i, I_{in}^i, t) = I_t^i - \alpha_t I_{res}^{\theta_i} + \delta_t I_{in}^i - \frac{\beta_t^2}{\bar{\beta}_t} \epsilon^\theta$ and the variance $\sigma_t = \frac{\beta_t \bar{\beta}_{t-1}}{\bar{\beta}_t}$. The variable $I_{res}^{\theta_i}$ is predicted using the generic UNet network. Through an implicit sampling strategy and reparameterization technique, $I_{t-1}^i$ can be sampled from $I_t^i$ as follows:

$$I_{t-1}^i = I_t^i - \alpha_t I_{res}^{\theta_i} + \delta_t I_{in}^i, \tag{14}$$

where the residual estimation value $I_{res}^{\theta_i}$ for each modality is optimized by the following objective:

$$L_{res}^i(\theta) = \mathbb{E}_{t, I_t^i, I_{res}^i, I_{in}^i}[\|I_{res}^i - I_{res}^{\theta_i}(I_t^i, I_{in}^i, t)\|_1]. \tag{15}$$

Shared-prior mechanism unifies heterogeneous degradations into a consistent latent mapping, ensuring physically plausible, controllable, and semantically complementary fusion across modalities.

## 3.2 DEGRADE FUSION MODULE

To operationalize the controllable and explainable diffusion dynamics, we design a degrade-fusion module to directly process multi-modal images from their degraded states, embedding diffusion priors without reconstructing from Gaussian noise. As shown in Fig. 2, this module consists of three key components:

### 3.2.1 DUAL-STREAM ENCODER

This initial component processes two pairs of inputs: $(I_{in}^a, I_{res}^a)$ and $(I_{in}^b, I_{res}^b)$, which respectively represent degraded multi-modal images and their predicted residuals relative to clean images. Following the computation of $I_{in}^i - I_{res}^i$ for each input pair, it is then directed through a dedicated four-level processing path (L1-L4). Within each level, features undergo processing by a block integrating a Transposed Self-Attention Block (TSAB) and a Spatial Self-Attention Block (SSAB). Progressing from shallower to deeper levels, the spatial dimensions of the feature maps are progressively downsampled, while their channel dimensions are concurrently expanded. The outputs from each level of both modality A and modality B pipelines are subsequently forwarded to the fusion component.

Figure 2: The architecture of our degrade-fusion module.

### 3.2.2 CROSS ATTENTION BLOCK

To effectively leverage the diffusion prior and establish robust inter-modal connectivity, we employed a cross-attention mechanism within the fusion module. This approach facilitated the fusion of features from both modalities, enabling the model to discern inter-modal interactions and correlations and thereby enhancing their respective representational capabilities, ultimately yielding fused features robustly informed by the diffusion prior.

### 3.2.3 DECODER AND PROGRESSIVE REFINEMENT

The decoder reconstructs the fused image via hierarchical upsampling from L4 to the original resolution. To preserve details, skip connections concatenate upsampled deeper features with corresponding shallower fused ones, with a final $L_r$ block refining the output $I_f$.

### 3.3 LOSS FUNCTION

The loss function largely determines whether the fusion result can effectively preserve discriminative information from multi-modal sources. Here, we construct an intensity loss $\mathcal{L}_{int}$ and a gradient loss $\mathcal{L}_{grad}$ to emphasize the preservation of salient contrast and rich textures, and enforce color consistency in the YCbCr space through a color loss $\mathcal{L}_{color}$ to maintain scene colors. The definitions of $\mathcal{L}_{int}$, $\mathcal{L}_{grad}$, and $\mathcal{L}_{color}$ are as follows:

$$\mathcal{L}_{int} = \frac{1}{HW}\|I_f - max(I_{vi}, I_{ir})\|_1. \tag{16}$$

$$\mathcal{L}_{grad} = \frac{1}{HW}\|\nabla I_f - max(\nabla I_{vi}, \nabla I_{ir})\|_1. \tag{17}$$

$$\mathcal{L}_{color} = \frac{1}{HW}\|\mathcal{F}_{CbCr}(I_f) - \mathcal{F}_{CbCr}(I_{vi})\|_1, \tag{18}$$

where $\{I_{ir}, I_{vis}\}$ denotes the infrared-visible image pair, $max(\cdot)$ is the maximum selection for preserving salient targets and textures, $\|\cdot\|_1$ represents the $l_1$-norm, $\nabla$ is the Sobel operator, and $\mathcal{F}_{CbCr}(\cdot)$ converts RGB to CbCr.

Finally, the total loss of our CLED-Fusion is formulated as the weighted sum of the aforementioned losses:

$$\mathcal{L}_{total} = \lambda_{int}\mathcal{L}_{int} + \lambda_{grad}\mathcal{L}_{grad} + \lambda_{color}\mathcal{L}_{color}, \qquad (19)$$

where $\lambda_{int}$, $\lambda_{grad}$, and $\lambda_{color}$ are hyperparameters for balancing various losses.

## 4 EXPERIMENTS

### 4.1 CONFIGURATIONS AND IMPLEMENTATION DETAILS

#### 4.1.1 DATASETS AND IMPLEMENTATION DETAILS

For degradation removal capabilities, our method employs the EMS Yi et al. (2024b) dataset, which features diverse explicit degradations (*e.g.*, low light, overexposure, rain, haze, blur, random noise in visible; low contrast, stripe noise, random noise in infrared). Fusion performance training utilizes the M3FD Tang et al. (2022a) dataset. Generalization is validated using LLVIP Jia et al. (2021) and Harvard medical datasets [1]. Experimental setup includes 3,344 training and 293 testing image pairs.

All experiments were conducted on a single NVIDIA RTX 3090 GPU using the PyTorch framework. In the first training stage, the model was trained on the EMS dataset for 300k iterations with the Adam optimizer, a learning rate of $8\times10^{-5}$, and a batch size of 8 (2 samples for low-light, 1 for other degradation types). The second stage involved training on the M3FD dataset for 300k iterations with a batch size of 4.

#### 4.1.2 EVALUATION METRICS AND BASELINES

Fusion results were evaluated using both quantitative and qualitative methodologies. Quantitative assessment employed metrics such as MI, VIF, $Q^{AB/F}$, $N^{AB/F}$, and SSIM, covering information content, visual perception, structural preservation, edge details, and consistency. This included both full-reference (*e.g.*, SSIM, VIF) and no-reference (*e.g.*, MI, $Q^{AB/F}$, $N^{AB/F}$) metrics for comprehensive evaluation.

Our method is benchmarked against several state-of-the-art competitors: Dif-Fusion Yue et al. (2023), VDMUFusion Shi et al. (2024), DRMF Tang et al. (2024), CCF Cao et al. (2024), Text-DiFuse Zhang et al. (2024), and GIFNet Cheng et al. (2025). Most of them are diffusion-based multi-modal fusion methods, with Text-DiFuse and DRMF addressing degraded scenarios.

### 4.2 COMPARATIVE EXPERIMENTS

#### 4.2.1 BENCHMARKING IMAGE FUSION METHODS IN REAL-WORLD SCENARIOS

Table 1: The quantitative comparison of CLED-Fusion with state-of-the-art image fusion methods on the M3FD and LLVIP datasets. (**Bold**:best result, *italic*:second result.)

| Methods | MI↑ | | VIF↑ | | $Q^{AB/F}$↑ | | $N^{AB/F}$↓ | | SSIM↑ | |
|---|---|---|---|---|---|---|---|---|---|---|
| | M3FD | LLVIP | M3FD | LLVIP | M3FD | LLVIP | M3FD | LLVIP | M3FD | LLVIP |
| Dif-Fusion(TIP'23) | *2.947* | **3.329** | 0.531 | 0.757 | 0.412 | 0.343 | 0.040 | 0.045 | 0.659 | 0.542 |
| VDMUFusion(TIP'24) | 2.633 | 2.794 | 0.569 | 0.748 | 0.309 | 0.340 | **0.003** | **0.001** | *0.891* | **0.763** |
| CCF(NIPS'24) | 2.714 | *2.916* | 0.493 | 0.779 | 0.253 | 0.344 | **0.003** | *0.009* | 0.777 | 0.694 |
| Text-DiFuse(NIPS'24) | 2.376 | 2.146 | 0.265 | 0.769 | 0.205 | 0.303 | 0.047 | 0.046 | 0.277 | 0.393 |
| GIFNet(CVPR'25) | 2.547 | 2.329 | *0.613* | *0.804* | *0.462* | **0.403** | 0.087 | 0.066 | 0.880 | *0.720* |
| CLED-Fusion(Ours) | **4.012** | 2.597 | **0.730** | **1.076** | **0.493** | *0.347* | *0.006* | 0.082 | **0.903** | 0.448 |

Fig. 3 visually presents real-world fusion performance on M3FD and LLVIP datasets, using captured images. CLED-Fusion consistently produces high-quality fused images even under challenging

---

[1] http://www.med.harvard.edu/AANLIB/home.html

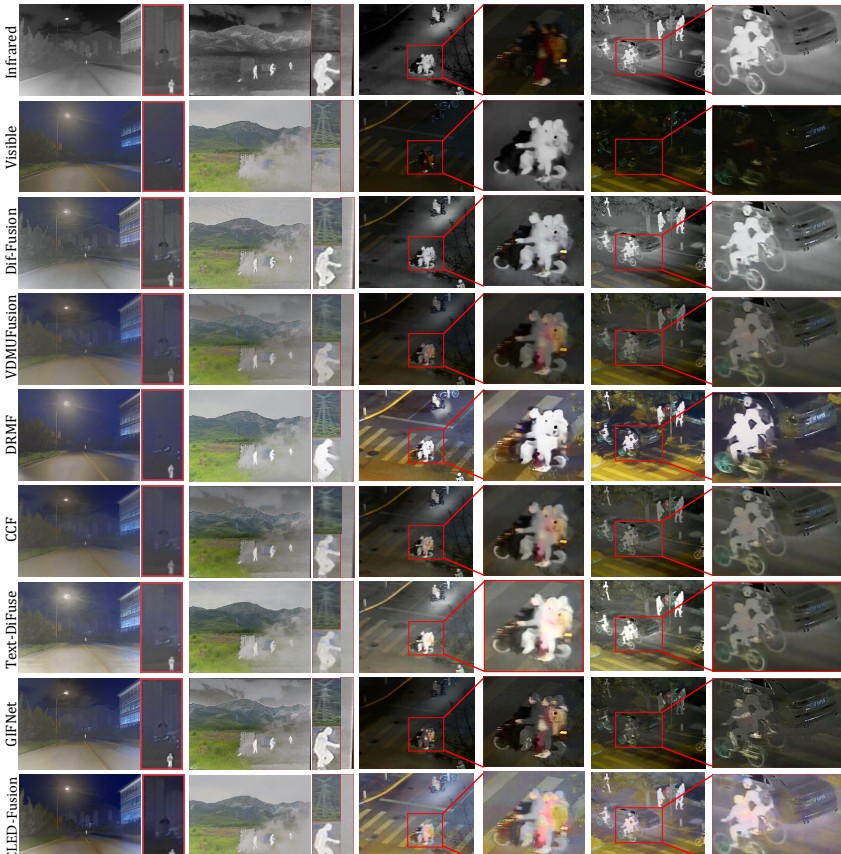

Figure 3: Visual comparison of CLED-Fusion with state-of-the-art image fusion methods on the M$^3$FD and LLVIP datasets.

nighttime and real-world weather conditions. While DRMF enhances quality for degraded images, it compromises detail, particularly in infrared content (e.g., background buildings, human details). Text-DiFuse achieves a balanced outcome but struggles to fully mitigate degradation interference from visible-light images. Competitors generally show limitations in robustly addressing degraded source images, often attenuating salient targets. Qualitative analysis indicates that diffusion-based fusion methods with degradation robustness enhance image quality and partially mitigate misregistration artifacts, as exemplified by CLED-Fusion, DRMF, and Text-DiFuse successfully eliminating black edges caused by misaligned source images (second column). This highlights a potential for future work to broaden degradation considerations to include spatial misalignment. Quantitatively, Table 1 corroborates our method exhibits highly competitive performance against current SOTA approaches across most metrics. Specifically, a superior VIF score validates enhanced suitability of fusion results for human visual perception, while a higher $Q^{AB/F}$ value signifies improved edge and detail preservation. This quantitative advantage aligns with observed performance of CLED-Fusion in real low-light environments, as shown in last two columns of Fig. 3.

### 4.2.2 ROBUSTNESS EVALUATION UNDER VARIOUS DEGRADATION CONDITIONS

To evaluate performance across diverse degradation types, we use the EMS dataset for real-world scenarios, including low contrast and noise in infrared, and noise, blur, overexposure, rain, haze in visible images. Futher, extensive testing on the LLVIP dataset (Fig. 3) focused on critical low-light conditions, prioritizing real-world relevance. Fig. 4 shows that methods designed for degradation, especially those in the last three rows, performed significantly better than approaches without such accounting. Our method distinguishes itself by accommodating the broadest range of degradations via a streamlined fusion process. Unlike DRMF, which needs pre-trained degradation models, or

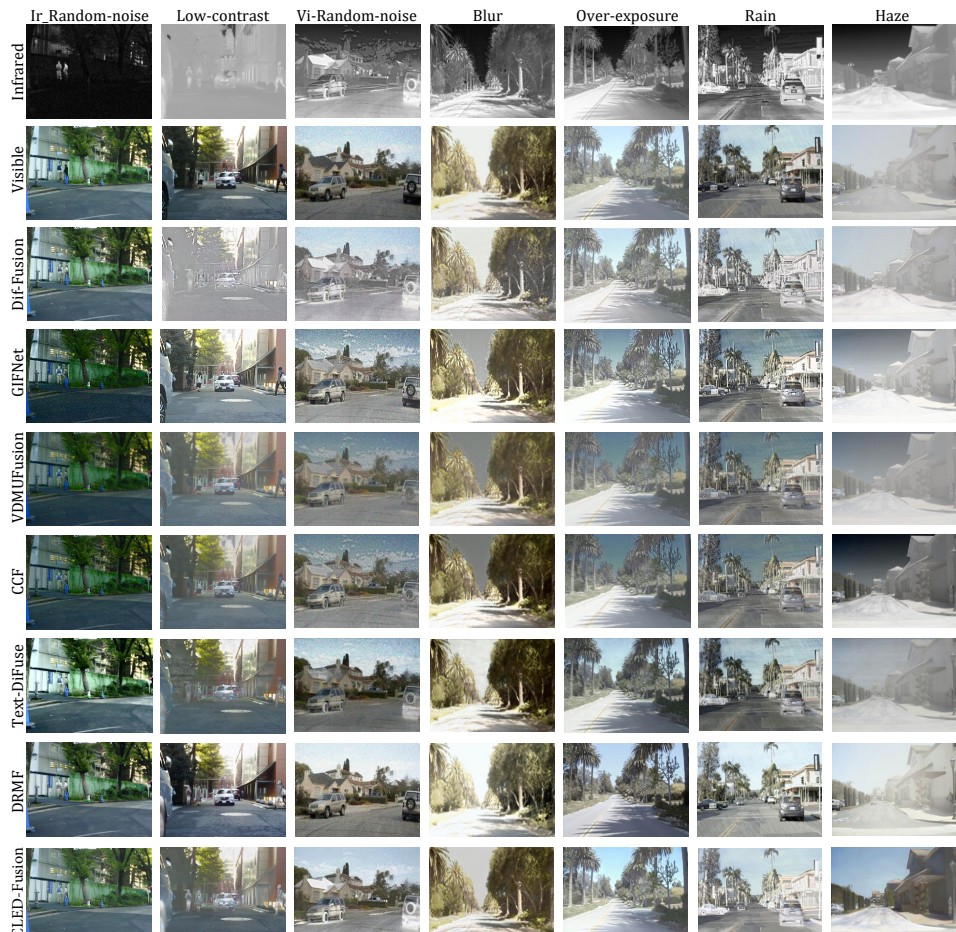

Figure 4: Visual comparison of CLED-Fusion's robustness with state-of-the-art image fusion methods under diverse degradation conditions on the EMS dataset.

Text-DiFuse, relying on user text instructions, our approach uses a single model to automatically detect and mitigate various infrared and visible-light degradations. This unified strategy provides a practical and scalable solution for complex real-world environments, where explicitly identifying degradation types is often infeasible. By jointly handling diverse degradations such as haze, rain, noise, blur, and low-light, our method demonstrates strong robustness and adaptability across modalities. Moreover, the framework's controllable design makes it readily extend to emerging degradation patterns, highlighting its potential for broader real-world applications.

## 5 CONCLUSION

A novel diffusion framework CLED-Fusion is proposed to explicitly integrate controllability and explainability into multi-degradation multi-modal image fusion. By enabling principled regulation of degradation removal and providing interpretable generative behavior, CLED-Fusion overcomes key limitations of existing approaches and establishes a new direction for robust and transparent multimodal perception in real-world scenarios.

While CLED-Fusion effectively addresses the challenges of controllability and explainability in multi-degradation multi-modal fusion, its current validation is primarily conducted on representative benchmarks and typical degradation types. Although sufficient to demonstrate the framework's generality and robustness, broader evaluations on more diverse modalities, dynamic real-world environments, and large-scale applications remain promising directions for future research.

## ETHICS STATEMENT

Our CLED-Fusion method significantly enhances the performance of infrared and visible light image fusion, making it extremely valuable for improving all-weather perception and decision-making in practical applications. Although the ability to directly fuse cross-modal imaging information brings potential risks, such as the introduction of artifacts or misleading features, we strongly recommend that researchers implement strict verification and supervision mechanisms to ensure the ethical use of these techniques. However, the original intention of image fusion is positive, aiming to promote more reliable perception in future multimodal systems. Therefore, we encourage researchers to utilize this technology with a responsible and cautious attitude.

## REPRODUCIBILITY STATEMENT

To ensure the reproducibility of our research results, detailed implementation instructions for CLED-Fusion can be found in the main text. In addition, the anonymous source code is available at `https://anonymous.4open.science/r/CLED-Fusion-D88C/`. These measures are designed to facilitate other researchers in the field to verify and reproduce our results.

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

# A APPENDIX

## A.1 EXTENSION EXPERIMENT

### A.1.1 OBJECT DETECTION

Table 2: Object detection performance comparison on the MSRS dataset.

| Detection | IR | VI | GIFNet | Text-DiFuse | CCF | VDMUFusion | DRMF | CLED-Fusion |
|---|---|---|---|---|---|---|---|---|
| mAP@0.5↑ | 71.9 | 74.8 | 94.5 | 89.7 | 95.0 | _95.3_ | 92.4 | **95.4** |
| mAP@[0.5:0.95]↑ | 48.4 | 47.3 | _72.7_ | 60.9 | 71.6 | 69.6 | 66.6 | **72.8** |

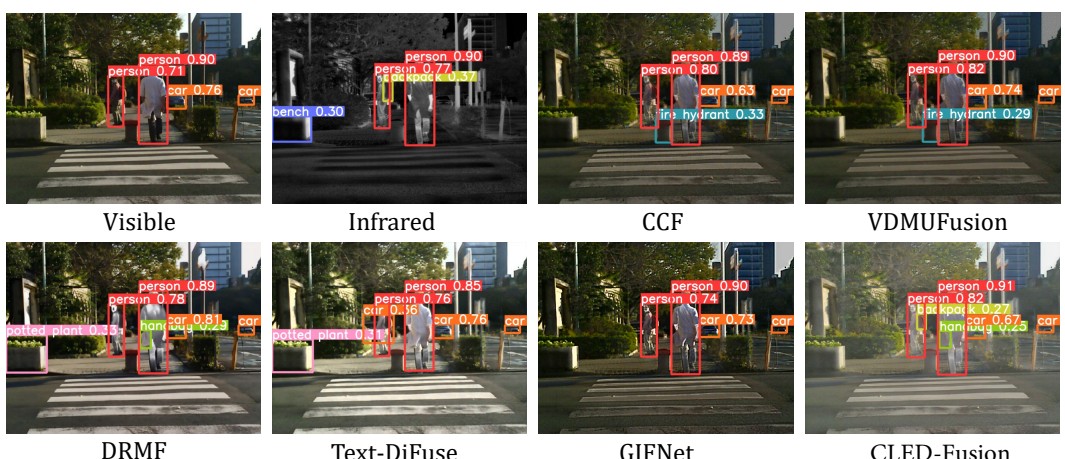

Figure 5: Visual of object detection performance comparison on MSRS dataset.

Effective information enhancement and aggregation not only aid in visual perception but also promote machine vision performance. To further demonstrate the superiority of DRMF, we evaluate its object detection performance on the MSRS dataset Tang et al. (2022b), employing a re-trained YOLOv9 Wang et al. (2024a) as the detector. As depicted in Fig. 5, CLED-Fusion successfully identifies all pedestrians with notably higher confidence scores. Table 2 shows CLED-Fusion achieves superior metrics, demonstrating its efficacy in preserving and enhancing information for downstream tasks.

### A.1.2 MEDICAL IMAGE FUSION

Table 3: Quantitative comparison in medical image fusion on Harvard dataset.

| Methods | MI↑ | VIF↑ | $Q^{AB/F}$↑ | $N^{AB/F}$↓ | SSIM↑ |
|---|---|---|---|---|---|
| Dif-Fusion(TIP'23) | 2.600 | 0.513 | 0.598 | 0.008 | 0.604 |
| VDMUFusion(TIP'24) | _3.291_ | 0.653 | 0.458 | 0.009 | _1.086_ |
| CCF(NIPS'24) | 2.731 | 0.563 | _0.625_ | _0.004_ | _0.635_ |
| Text-DiFuse(NIPS'24) | 2.843 | 0.579 | 0.205 | 0.018 | 0.534 |
| GIFNet(CVPR'25) | 2.581 | 0.413 | 0.345 | 0.030 | 0.443 |
| CLED-Fusion(Ours) | **3.924** | **0.939** | _0.739_ | _0.002_ | **1.260** |

Further generalization validation of CLED-Fusion was conducted on medical images, as detailed in Fig. 6 and Table 3. Our method consistently achieved significantly superior results compared to existing approaches. This robust performance further underscores the universality of CLED-Fusion, demonstrating its effective applicability across diverse domains, tasks, and datasets.

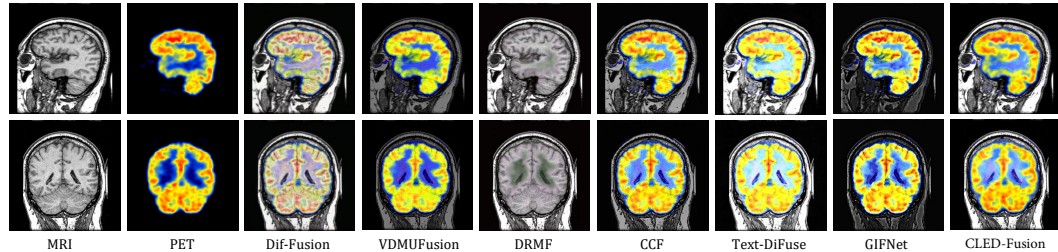

| MRI | PET | Dif-Fusion | VDMUFusion | DRMF | CCF | Text-DiFuse | GIFNet | CLED-Fusion |

Figure 6: Visual comparison in medical image fusion on Harvard datasets.

## A.2 ABLATION STUDIES

The ablation studies presented in Fig 7 and Table 4 underscored the essential contributions of the various loss functions and the trianing strategy. Specifically, the omission of $L_{int}$ rendered the generation of valid fused images impossible, whereas the absence of $L_{grad}$ significantly compromised edge and detail preservation. $L_{color}$ was found to be indispensable for maintaining color fidelity. Moreover, experiments with single-stage model training demonstrated a degradation of low-frequency information generation, particularly on low-light scene datasets, highlighting a key challenge for future optimization.

Table 4: The ablation studies of CLED-Fusion on M³FD and LLVIP datasets.

| Method | M³FD | | | | | LLVIP | | | | |
|---|---|---|---|---|---|---|---|---|---|---|
| | MI↑ | VIF↑ | $Q^{AB/F}$↑ | $N^{AB/F}$↓ | SSIM↑ | MI↑ | VIF↑ | $Q^{AB/F}$↑ | $N^{AB/F}$↓ | SSIM↑ |
| w/o $L_{int}$ | 0.459 | 0.057 | 0.102 | 0.000 | 0.004 | 0.255 | 0.126 | 0.127 | 0.021 | 0.002 |
| w/o $L_{grad}$ | 3.420 | 0.663 | 0.451 | 0.014 | 0.803 | 2.683 | 1.325 | 0.317 | 0.151 | 0.412 |
| w/o $L_{color}$ | 3.324 | 0.670 | 0.455 | 0.007 | 0.856 | 2.666 | 1.242 | 0.353 | 0.102 | 0.463 |
| one-stage | 3.720 | 0.715 | 0.436 | 0.030 | 0.850 | 1.733 | 0.960 | 0.120 | 0.346 | 0.079 |
| CLED-Fusion | 4.012 | 0.730 | 0.493 | 0.006 | 0.903 | 2.597 | 1.076 | 0.347 | 0.082 | 0.448 |

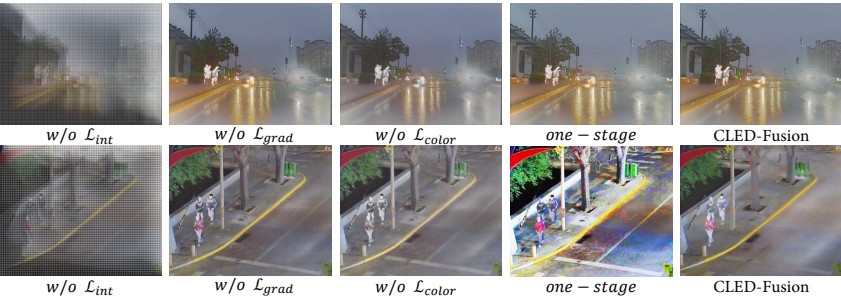

Figure 7: The visual of ablation studies.

