# OpenReview forum: "CLED-Fusion: Controllable and Latent-Explainable Diffusion for Multi-Degradation Multi-Modal Image Fusion"
_ICLR.cc/2026/Conference — ICLR 2026 Conference Withdrawn Submission_

### Official Review · Reviewer_NYDU · 2025-10-30

**Soundness:** 2
**Presentation:** 3
**Contribution:** 2
**Rating:** 4
**Confidence:** 5

**Summary:**

This paper proposes CLED-Fusion, a diffusion-based framework for multi-degradation multi-modal image fusion. The authors aim to enhance controllability and explainability by introducing:
- a dual-path diffusion process that separates deterministic residual degradation from stochastic noise;
- a shared prior mechanism to unify heterogeneous degradations from different modalities;
- a degrade-fusion module that directly embeds diffusion priors into degraded inputs without reconstructing from Gaussian noise.
Experiments on M3FD, LLVIP, and Harvard datasets reportedly show improved robustness and fusion quality under various degradations.

**Strengths:**

- Ambitious attempt to address both controllability and explainability in diffusion-based fusion.
- Clear mathematical formulation and step-by-step derivation of the dual diffusion process.
- Comprehensive set of datasets and baseline methods for comparison.

**Weaknesses:**

- The proposed residual-noise diffusion decomposition is conceptually similar to ResShift [1] and RDDM [2], without highlighting substantial innovation.
- The paper does not justify why a computationally heavy diffusion model is required for “degradation prior” extraction rather than simpler regression-based priors.
- Inputs I_res^a and I_res^b appear in formulas but are never introduced earlier in the paper.
- The loss function uses the degraded source images (I_ir, I_vis) as references, while Fig. 2 shows that the loss inputs are the degraded images themselves.
- Missing baseline. Despite using EMS from Text-IF, the method does not compare directly to Text-IF, weakening the claimed performance advantage.
- In Figure 5, the fusion results exhibit severe infrared dominance. In Figure 6, the proposed method fails to maintain fine structural details of MRI images.
- It is well known that diffusion models are computationally intensive. However, the authors do not provide any statistics on model parameters, FLOPs, or average runtime, making it difficult to assess the practical feasibility and efficiency of the proposed method.
- The citation of M3FD dataset (line 339) is incorrect.
- In the degradation experiments, pre-enhancement should be used to ensure fair comparison among methods focusing only on fusion.

[1] Yue Z, Wang J, Loy C C. Resshift: Efficient diffusion model for image super-resolution by residual shifting[J]. Advances in Neural Information Processing Systems, 2023, 36: 13294-13307.
[2] Liu J, Wang Q, Fan H, et al. Residual denoising diffusion models[C]//Proceedings of the IEEE/CVF Conference on Computer Vision and Pattern Recognition. 2024: 2773-2783.

**Questions:**

- Can the first-stage diffusion model directly output restored clean images, thus serving as a pre-processing step for other fusion methods?
- How were I_res^a and I_res^b obtained?
- Please report computational metrics (parameters, FLOPs, average inference time).

---

### Official Review · Reviewer_Y8i3 · 2025-10-30

**Soundness:** 2
**Presentation:** 3
**Contribution:** 3
**Rating:** 4
**Confidence:** 3

**Summary:**

This paper introduces CLED-Fusion, a novel Controllable and Latent-Explainable Diffusion framework for multi-modal image fusion under diverse degradations. Unlike existing methods that focus on feature integration, CLED-Fusion offers controllability over the degradation removal process and explainability of its generative steps.

**Strengths:**

1.The overall structure and organization of the paper are well-designed and logically presented.

2.The motivation is clearly stated and easy to follow, effectively highlighting the significance of the work.

3.The figures are visually appealing, with color schemes that enhance readability and overall presentation quality.

**Weaknesses:**

- The description of previous methods in the Related Work section, particularly regarding their focus on information integration, is not sufficiently clear. As a result, the discussion does not effectively highlight or contrast the innovations and advantages of the proposed approach.
- The authors do not show how α&β is decided precisely during training (in equation 4, 11).
- The authors claim to introduce shared distribution priors into a consistent latent space; however, it remains unclear what these priors specifically represent and how they are derived or learned.
- The authors state that CLED-Fusion enables cross-modal balance, yet it is not clearly explained how this balance is achieved. The paper lacks a concrete description or mechanism demonstrating how the method modulates or controls such balance in practice.
- It is unclear what the core distinction between TSAB and SSAB is. Is the difference merely in the feature dimension on which they operate, or do they serve fundamentally different functional roles? Further clarification on their respective mechanisms and contributions would strengthen the paper.

**Questions:**

See Weaknesses.

---

### Official Review · Reviewer_mMHK · 2025-11-01

**Soundness:** 2
**Presentation:** 2
**Contribution:** 2
**Rating:** 4
**Confidence:** 4

**Summary:**

The paper introduces a image fusion framework called CLED-Fusion, which addresses the challenges of controllability and explainability in multi-degradation, multi-modal image fusion. The approach reformulates diffusion dynamics into a dual pathway: a deterministic residual path for degradation removal and a stochastic noise path for detail preservation. It incorporates shared distribution priors to unify heterogeneous degradations across modalities into a consistent latent space, and an explicit degrade-fusion module to integrate inputs efficiently without full reconstruction from Gaussian noise.

**Strengths:**

1. The introduction of a dual-pathway diffusion process (residual and noise pathways) helps disentangle degradation factors and fine-grained details.
2. By embedding priors directly into degraded inputs and using few denoising steps, the framework avoids the computational overhead of traditional diffusion-based fusion.
3. Experiments cover diverse benchmarks, including quantitative metrics and qualitative visuals. The paper demonstrates robustness to unseen degradations and benefits for downstream tasks like object detection.

**Weaknesses:**

1. The empirical depth on controllability and explainability is limited. While controllability and explainability are key claims, the experiments primarily show the quantitative comparison of SOTA models and qualitative examples of varying degradation. The experimental data on controllability is very limited. Furthermore, the demonstrations of explainability are restricted to trajectory visualizations in Fig. 1, with no formal analysis of explainability provided.

2. The authors mention that the degradation handling capabilities of existing methods are design-dependent, resulting in insufficient extensibility. Please explain how the proposed method improves extensibility compared to previous methods, and what specific design elements contribute to this enhancement.

3. The ablation studies are insufficient. The authors only perform ablation on the three loss functions in the supplementary materials, lacking ablation experiments on key modules such as the residual-noise decomposition and shared prior mapping. This limits the ability to substantiate the effectiveness of the proposed contributions.

4. There is a lack of detailed explanation regarding the specific training process and setup. How was the model trained? What training datasets were used? Is the comparison with the baseline methods fair? A more detailed explanation is needed.

5. The authors mention in Section 4.1.2 "Evaluation Metrics and Baselines" that a comparative experiment with DRMF[1] should be conducted. However, no performance comparison with DRMF is presented in Tables 1, 2, or 3. Given that DRMF is a closely related work, it is recommended to include the relevant comparative experimental results.
[1] Tang L, Deng Y, Yi X, Yan Q, Yuan Y, Ma J. DRMF: Degradation-robust multi-modal image fusion via composable diffusion prior. InProceedings of the 32nd ACM International Conference on Multimedia 2024 Oct 28 (pp. 8546-8555).

**Questions:**

See the weakness

---

### Official Review · Reviewer_Qitj · 2025-11-01

**Soundness:** 2
**Presentation:** 1
**Contribution:** 2
**Rating:** 2
**Confidence:** 5

**Summary:**

This paper introduces CLED-Fusion, a diffusion-based framework for multi-modal image fusion under diverse degradations such as low-light, blur, haze, and noise. The approach is designed with controllability and interpretability as central objectives by disentangling the generative process into a deterministic residual pathway to explicitly model degradations and a stochastic noise pathway for uncertainty and fine detail. A key contribution is the introduction of shared distribution priors to unify degradations from different modalities into a consistent latent space, enabling explicit control over the strength of degradation removal and cross-modal balance. Extensive experiments on public benchmarks (including EMS, M3FD, LLVIP, and medical datasets) indicate CLED-Fusion delivers strong performance, robustness to various degradation types, and valuable generalization to practical scenarios, including medical image fusion.

**Strengths:**

1. The introduction of shared distribution priors (Eqns. in Section 3.1.2) and a selective hourglass mapping allows for the alignment of heterogeneous degradations across modalities.
3.  The paper contextualizes its gaps with respect to recent diffusion-based fusion works, providing solid discussion on where previous methods (DRMF, Text-DiFuse, etc.) fall short regarding generality, scalability, and interpretability.

**Weaknesses:**

**1. The organization of the paper is unsatisfactory.**
	a. The results in Section 4.2.2 lack any quantitative comparison.
	b. The entire experimental section contains no ablation studies, and the experiments on *medical imaging* mentioned in the abstract do not appear in the main text. I understand that some results are placed in the supplementary materials; however, the main paper itself is not self-contained. Under a page-limit constraint, this presentation is unfair compared with other submissions.

**2. The paper’s contributions are not clearly articulated.**
	a. After carefully reading the *Introduction* and *Method* sections, I still cannot clearly understand where the claimed *controllability* and *explainability* are reflected. Furthermore, for the proposed *shared distribution priors* and *deterministic residual pathway*, I remain confused about the motivation behind these designs and how they specifically contribute to improving controllability and explainability in image fusion.
	b. In particular, there is no convincing experimental support (e.g., ablation or empirical studies) demonstrating the effectiveness of these claimed contributions.

**3. Concerning the experimental results:**
	a. From the qualitative results in Figure 3, I do not observe clear advantages of your method; in some regions, the results even appear blurrier. In Figure 5, DRMF and Text-DiFuse clearly produce better fusion outcomes.
	b. The paper lacks comparisons with several relevant methods that are mentioned in the related work section, such as CDDFuse, DDFM, Diff-IF. Moreover, DCEvo [1].
	[1] DCEvo: Discriminative Cross-Dimensional Evolutionary Learning for Infrared and Visible Image Fusion, CVPR 2025.
	c. Given such a complex pipeline design, there is *no quantitative benchmarking* of computational cost, training time, or memory usage. These results are essential for assessing the method’s applicability in real-time or resource-constrained settings.
	d. The ablation study descriptions and analyses are overly brief and lack evaluations on the key proposed components — namely, the *deterministic residual pathway*, *shared distribution priors*, and *explicit degrade-fusion module*.
        e. Section 4.2.2 lacks quantitative results, making it difficult to quantitatively appreciate the superiority of the method. Additionally, when performing fusion comparisons on degraded images, I recommend equipping the baseline methods with a jointly-trained or post-training image enhancement module, to demonstrate that your approach can better accommodate multi-degradation. Otherwise, the comparison is unfair, since some competing methods were only trained on clear images.

**4. Additional details:**
	a. Section 3.1.2 lacks clarity in defining the roles of the coefficients $\alpha_t$, $\delta_t$, and $\beta_t$ within the residual and shared-distribution terms. The derivation of the reverse-process mean $u_\theta$ is terse and should better justify how these coefficients are combined and interact.
	b. Line 048 (“and diffusion models…”) lacks citations.
	c. The two-stage training procedure (Section 4.1.1) is not critically analyzed for potential overfitting to specific degradations. Its impact on generalization to unseen conditions remains unquantified beyond qualitative observations.

**Questions:**

Detailed issues and critiques are provided in the Weaknesses section.

---

### Note · Authors · 2025-11-18

I have read and agree with the venue's withdrawal policy on behalf of myself and my co-authors.